# Aligning time series anomaly detection research with practical applications

**Daniel Barrish**                                                     *daniel.barrish@gmail.com*
*Department of Industrial Engineering*
*Stellenbosch University*

**Jan van Vuuren**                                                     *vuuren@sun.ac.za*
*Department of Industrial Engineering*
*Stellenbosch University*

**Reviewed on OpenReview:** *https://openreview.net/forum?id=RyMLAr5tFU*

## Abstract

The field of time series anomaly detection is hindered not by its models and algorithms, but rather by its inadequate evaluation methodologies. A growing number of researchers have claimed in recent years that various prevalent metrics, datasets, and benchmarking practices employed in the literature are flawed. In this paper, we echo this sentiment by demonstrating that widespread metrics are incongruent with desirable model behaviour in practice and that datasets are plagued by inaccurate labels and unrealistic anomaly density, amongst other issues. Furthermore, we provide suggestions and guidance on realigning theoretical research with the demands of practical applications, with the goal of establishing a stable, principled benchmarking framework within which models may be evaluated and compared fairly. Finally, we offer a perspective on the main challenges and unanswered questions in the field, alongside potential future research directions.

## 1    Introduction

The field of time series anomaly detection has many direct applications in wide-ranging domains such as finance, healthcare, and manufacturing. Despite this, the theoretical applications and real-world use cases are often misaligned as a result of unsuitable anomaly detection metrics and unreliable benchmarking datasets. Since these flawed conventions have become ingrained in the literature, many researchers are regrettably building their castles on sand.

We suggest re-examining these fundamental building blocks of time series anomaly detection, adopting an application-driven approach in order to establish a principled set of benchmarking practices. After providing a brief introduction to anomaly detection in time series and outlining previous criticisms of the field in §2, we discuss some typical applications in §3 and formulate a set of underlying assumptions that they all share. This is followed by a critique of established anomaly detection metrics, leading to empirical results demonstrating the efficacy of two recent metrics in §4. We then consider the issue of datasets in §5, which includes a list of desirable characteristics, a critique of prominent public datasets, and a few suggestions on how to proceed with developing new datasets. Thereafter, the state of anomaly detection models is discussed in §6, as is the need for sensible baselines and our broad suggestions for future research directions. Finally, we conclude in §7 with a summary of the main points and suggested follow-up research avenues.

## 2    Related work

In this section, we provide brief background information on anomaly detection in time series, as well as a review of some of the recent criticisms facing the field.

### 2.1 Time series anomaly detection approaches

Time series anomaly detection is the process of identifying observations or subsequences within a time-ordered sequence of data that deviate significantly from the expected behaviour or underlying patterns of the series, typically under the assumption that such deviations are rare and represent events of interest. Anomaly detection is usually seen as a process consisting of two steps: anomaly scoring and thresholding (Keogh, 2021). The former is concerned with determining a score for each timestep in the series—conventionally (although not necessarily) within a $[0, 1]$ range, with 0 representing the most normal data and 1 representing the most anomalous. Thereafter, the latter step involves converting these scores into binary predictions (i.e. normal or anomalous).

Since appropriate thresholds tend to be context-dependent, and the fact that anomaly scoring is often considered the more difficult step (Keogh, 2021), most research in the literature has focused on developing better scoring models. Many scoring approaches have been proposed, ranging from traditional statistical approaches (e.g. statistical process control charts or a moving average) to newer machine learning methods (e.g. isolation forest and local outlier factor) to the very latest deep learning techniques (e.g. autoencoders, transformers, and numerous other architectures). These models are not the focus of the present paper, and instead we refer the interested reader to one of the many comprehensive surveys on the current landscape of time series anomaly detection algorithms (Schmidl et al., 2022; Zamanzadeh Darban et al., 2024).

### 2.2 Previous criticisms of the field

In recent years, a growing body of research has supported the notion that novel algorithms in the field are often built on questionable foundations (Lai et al., 2021; Keogh, 2021; Hwang et al., 2022; Kim et al., 2022; Wu & Keogh, 2023; Liu & Paparrizos, 2024; Sarfraz et al., 2024; Sørbø & Ruocco, 2024). Without insightful metrics, robust datasets, appropriate baselines, and sound benchmarking practices, any results or conclusions are unreliable at best. An increasing number of dissenting researchers have raised concerns about the direction of the field, with the general sentiment being that more research effort should be devoted to establishing a solid foundation for benchmarking (i.e. metrics, datasets, and baselines) before pursuing the development of more advanced and capable models (Sarfraz et al., 2024).

## 3 Time series anomaly detection applications

In this section, we review and discuss how time series anomaly detection is currently applied. Thereafter, we identify certain commonalities in these applications and present a set of assumptions that applies to most use cases, which subsequently inform our criticisms and suggestions pertaining to metrics, datasets, and models in later sections.

### 3.1 Typical use cases

Considering the generic and fundamental nature of time series anomaly detection, it is unsurprising that it is employed in a wide variety of domains. Some of the most prominent applications include:

- **Healthcare.** A simple application is monitoring a patient's vital signs (such as heart rate, blood pressure, etc.) using sensors in a bid to pre-empt any potential medical emergency (Haahr-Raunkjaer et al., 2022). More complex applications include the analysis of electrocardiograms aimed at detecting heart arrhythmias (Greenwald et al., 1992; Moody & Mark, 2001) and employing electroencephalograms for the identification of unusual brain activity that might precede an epileptic seizure (Shoji et al., 2021).

- **Predictive maintenance.** One of the field's most prominent use cases is monitoring key sensor readings from machinery or equipment (such as temperature, vibration, and pressure) in order to pre-empt catastrophic failures by carrying out predictive maintenance before a breakdown occurs (Choi et al., 2022; Barrish & van Vuuren, 2023). Another potential application is in quality assurance, as deviations from normal system behaviour might result in defective products.

- **Finance and economics.** Time series anomaly detection frequently plays a role in credit card fraud detection systems by monitoring spending habits (Moschini et al., 2021). It may also be employed to aid algorithmic trading strategies or detect illegal trading (James et al., 2023).

- **Information technology operations and cybersecurity.** Anomaly detection models are widely used to aid in network intrusion detection by analysing network traffic (García-Teodoro et al., 2009). Unusual patterns in packet sizes or connection frequency might, for instance, indicate that a cyberattack has occurred. Another use case is monitoring data generated from servers (such as CPU usage and disk I/O) with a view to flag any hardware failures or software bugs early (Su et al., 2019).

### 3.2 Common assumptions

Despite the diversity of these applications, anomaly detection is employed in fundamentally similar ways. In each case, a data-generating process or system (e.g. an electrocardiogram, machine temperature, or network traffic) is analysed or monitored automatically by some model. If an abnormality in the data is identified, a domain expert (e.g. cardiologist, technician, or information technology specialist) is alerted to the existence of a potential anomaly. Based on a closer inspection of the underlying process, the expert decides whether the issue is genuine and severe enough to warrant further action. Anomaly detection may be seen as a form of decision support since it aids experts in deciding where they should focus their attention—this human factor is an important consideration when attempting to align theory with practice.

Based on the typical applications described in §3.1 and the workflow outlined above, the following reasonable assumptions inform our suggestions with respect to metrics and datasets:

- Labelled, ground truth anomalies correspond to events which are *noteworthy* to the user of the anomaly detection system. Anomalies are somewhat subjective by definition, but what we are assuming here is that system users know whether an alert is useful or not, and this knowledge may be leveraged to determine whether similar alerts should be issued in the future.

- Experts investigate each alert soon after the potential anomaly is flagged. Within a reasonable amount of time, these experts can determine whether a true anomaly was flagged based on their subjective interpretation of "usefulness." One flag during an anomalous subsequence is sufficient to uncover the entire anomaly, and subsequent alerts are redundant.

- Users of the anomaly detection system are primarily concerned with the system's ability to detect anomalies. The number of true negatives should not play a role (or at least not a significant role) in the evaluation of anomaly detection models, as the imbalanced nature of anomaly detection datasets may result in overly optimistic evaluations.

Naturally, this is not an exhaustive list of possible assumptions covering all situations—various use cases will often have more specific requirements. For instance, in many applications such as high-frequency trading, timeliness of the detection is important. In such cases, earlier detections should be rewarded, but the exact degree to which it should be rewarded varies significantly and thus cannot easily be accounted for in general terms. Similarly, in certain applications (such as the monitoring of machinery in an industrial setting), the duration of an anomaly (such as overheating, for instance) may correlate with the severity. Without additional contextual information, it is also impossible to account for this in generic benchmarks.

Although these assumptions do not apply universally, the point of clearly stating these here is to establish a basic framework that dictates what anomalies are and how they are handled in practice, particularly within human-in-the-loop monitoring contexts. This framework may then be utilised when selecting or designing metrics as well as guiding the curation of datasets.

## 4 Metric alignment

The importance of selecting suitable performance metrics is often underestimated, especially in time series anomaly detection where the selection of appropriate metrics is particularly challenging due to the temporal

nature of the data. Many researchers have opted for intuitive measures borrowed from tabular anomaly detection or binary classification, but often each of these has at least one serious drawback. A widespread failure to assess evaluation approaches critically prior to their use has resulted in flawed metrics permeating the literature. We discuss the most prominent of these metrics in this section, before presenting our recommendations.

### 4.1 A critique of existing metrics

Metrics may be partitioned into two broad categories: those in which the final binary predictions are compared with the labels, and those that involve evaluating the anomaly scoring approach in isolation without thresholding. Sørbø & Ruocco (2024) refer to these as *binary* (threshold-specific) and *non-binary* (threshold-free) metrics, respectively.

Let us examine binary metrics first, which are typically defined as some combination of true positives (TPs), false positives (FPs), true negatives (TNs), and false negatives (FNs). Within this category we have:

- **Accuracy** is defined as $A = (TP_P + TN_P)/n$ where $n$ is the length of the time series, $TP_P$ is the number of true positive points, and $TN_P$ is the number of true negative points. This is a particularly poor metric due to the imbalanced nature of anomaly detection datasets. As a result, models may achieve a high score by flagging no anomalies in the time series (Sørbø & Ruocco, 2024). Fortunately, it has not found widespread use in the time series anomaly detection literature.

- **Point-wise recall** is defined as $R_P = TP_P/(TP_P + FN_P)$, and may be interpreted as the fraction of actual anomalous points that were flagged correctly. Since this incentivises models to flag everything due to no penalty being imposed for false positives, recall is complemented by the following metric (Tatbul et al., 2018).

- **Point-wise precision** is defined as $P_P = TP_P/(TP_P + FP_P)$, which may intuitively be seen as the fraction of flagged points that were actual anomalies. This rewards models for only flagging points when they are absolutely certain, which may result in many undetected anomalies (Tatbul et al., 2018).

- **The point-wise $F_1$ score** attempts to balance precision and recall. It is defined as the harmonic mean of the point-wise precision and point-wise recall, or $F_1 = (2 \times P_P \times R_P)/(P_P + R_P)$. It is an instance of the generic $F_\beta$ score, where $\beta$ is the additional weight afforded to recall relative to precision. While this metric seems sensible and has found use in at least 14 time series anomaly detection papers (Sørbø & Ruocco, 2024), we believe it is flawed since contiguous anomalous points are treated as *events* in practice as opposed to individual points (Garg et al., 2021). This results in partial detection of an anomalous sequence being penalised too harshly, which does not reflect desirable behaviour in practice (Xu et al., 2018; Sørbø & Ruocco, 2024).

- **The point-adjusted $F_1$ score** attempts to remedy the above flaw by treating all points in an anomalous subsequence as true positives if any point is flagged, thereby rewarding partial detection (Xu et al., 2018). After the adjustment, the $F_1$ score is calculated similarly, using the point-adjusted precision and recall: $F_{A,1} = (2 \times P_A \times R_A)/(P_A + R_A)$. It has seen widespread use in more than 16 papers (Sørbø & Ruocco, 2024), despite having been shown to result in overly optimistic evaluations due to the attainment of an artificially high precision value after the adjustment. Numerous studies have, in fact, demonstrated that random predictors are often able to outperform renowned anomaly detection models (Garg et al., 2021; Doshi et al., 2022). As a consequence, we contend that this metric should be abandoned in the context of time series anomaly detection.

- **The event-wise $F_1$ score.** When contiguous sequences of anomalous labels or anomalous predictions are considered a single event, one can redefine precision and recall as $P_E$ and $R_E$, respectively (Hundman et al., 2018). Furthermore, these may be used to define the event-wise (also known as the segment-wise) $F_1$ score as $F_{E,1} = (2 \times P_E \times R_E)/(P_E + R_E)$. This metric has been

employed at least five times in the literature despite the following fatal flaw: assuming that there is at least one anomaly in the time series, the model may achieve a perfect score by flagging the entire time series.

- **The composite $F_1$ score** is not well-known and combines event-wise recall with point-wise precision. It is defined as $F_{C,1} = (2 \times P_P \times R_E)/(P_P + R_E)$, which means that models are rewarded once for each detected anomalous event and penalised repeatedly for each false positive point (Garg et al., 2021). Of the metrics discussed thus far, we believe this is the one most aligned with practice, but it still violates the second assumption outlined in §3.2 by rewarding redundant true positives during a single anomalous event.

In contrast with binary metrics, non-binary metrics deal with raw anomaly scores before a threshold is applied, typically with the aim of evaluating anomaly scoring methods in isolation. Prominent examples of such metrics include:

- **The area under the receiver operating characteristic curve** ($AUC_{ROC}$) is obtained by numerically integrating the receiver operating characteristic curve, which plots the true positive rate against the false positive rate. The true positive rate is simply another term for recall, whereas the false positive rate is defined as $FPR = FP/(FP + TN)$. The problem here is that the number of true negatives may positively skew the results, which should not be the case since true negatives are largely irrelevant in practice, as mentioned in §3.2. Despite this flaw, the $AUC_{ROC}$ has been employed more than 16 times in recent studies (Sørbø & Ruocco, 2024).

- **The area under the precision-recall curve** ($AUC_{PR}$) is obtained by numerically integrating the precision-recall curve. Since there is a trade-off between point-wise precision and recall at each fixed threshold, $AUC_{PR}$ provides a good measure of anomaly scoring performance across all thresholds. It features prominently in the literature (in at least nine time series anomaly detection papers (Sørbø & Ruocco, 2024)), and while we certainly believe it is a marked improvement over the $AUC_{ROC}$ score (Saito & Rehmsmeier, 2015), it suffers from the same flaws as the point-wise $F_1$ score.

- **The UCR (University of California, Riverside) score** (Keogh, 2021; Wu & Keogh, 2023) is defined as the precision with one prediction, or $P@1$, with an additional constraint stipulating that each problem in the dataset only contains a single anomaly. Since the need for a threshold is eliminated, anomaly scoring techniques may be evaluated in isolation by aggregating $P@1$ scores. Unfortunately, although we agree with the UCR score's philosophy, this metric has limited applicability since few benchmark datasets in practice have only one anomaly in each time series.

We note that other metrics have been proposed as well, such as the volume under the surface (Paparrizos et al., 2022a), the range-based $F_1$ score (Tatbul et al., 2018), and the Numenta anomaly benchmark score (Lavin & Ahmad, 2015). At the time of writing, however, these other metrics have been used relatively infrequently in the literature as they typically operate under different assumptions or introduce additional parameters which are designed for specific use cases.

More specifically, the mechanism underlying the metrics proposed by Paparrizos et al. (2022a), such as volume under the surface, still aggregates point-wise performance and rewards redundant true positives. On the other hand, the customisability of the range-based $F_1$ score (Tatbul et al., 2018) comes with the added complexity of a tunable weight and up to six tunable functions, making it ill-suited for benchmarking purposes. A detailed comparison between our recommended metrics and range-based approaches would be valuable future work, but was considered outside the scope of the current paper.

### 4.2 Suggested metrics

All of the metrics discussed above have at least one clear drawback. Fortunately, we believe that a simple alteration of the formula for precision so that it ignores redundant true positives (where the same anomalous event is flagged again) addresses these weak points. What Barrish (2025); Barrish & van Vuuren (2026) call the *realistic precision* is defined as $P_R = TP_E/(TP_E + FP_P)$.

This notion may be used to define our recommended binary metric, the *realistic $F_1$ score*, as $F_{R,1} = (2 \times P_R \times R_E)/(P_R + R_E)$. This metric is perfectly aligned with what we considered desirable behaviour: correctly detected anomalous events are rewarded once, subsequent redundant true positives are ignored, and each false flag is penalised individually. We obtain a principled non-binary metric by adapting the $AUC_{PR}$ score in a similar way in terms of $P_R$ and $R_E$, which now may be used to compare anomaly detection algorithms in respect of datasets that contain multiple anomalies.

Although these two metrics are aligned with the most basic, parameter-free assumptions in many applications, various use cases may have more specific requirements. Fortunately, both metrics may be extended easily to different contexts and applications—for instance, if one desires to account for different costs associated with false positives and false negatives then one may incorporate a $\beta$ value other than 1. Similarly, if detection timeliness is a factor, then one can easily enforce a minimum time within which each anomaly must be flagged.

We do not claim that these two metrics are a panacea for the problem of metrics in time series anomaly detection. Instead, we recommend these two options as satisfying the assumptions we set out in §3.2. It is certainly possible that there are other, better options, and we strongly encourage further research in this direction. Moreover, due to the complex and varied nature of time series anomaly detection, we believe that the choice of metric in papers should be considered carefully.

To validate empirically these claims, we conducted a systematic benchmarking study of six algorithms across three datasets. The results, detailed in Appendix A, demonstrate significant rank reversals when switching from standard to realistic metrics. Notably, we show that point-adjusted F-scores can award "state-of-the-art" performance to a purely random scorer, whereas the recommended realistic F-score correctly identifies it as noise.

## 5    Dataset alignment

For benchmarking results to be fair, robust, and reliable, it is essential that the datasets employed are of a high standard. Unfortunately, the dearth of high-quality datasets in the field is one of the largest challenges facing time series anomaly detection (Wu & Keogh, 2023). In this section, we first discuss some desirable characteristics of datasets. Thereafter, we briefly review some of the most prominent datasets in the literature, before providing our suggestions.

### 5.1    Desirable dataset characteristics

Since it is desirable to provide a good estimate of real-world performance, datasets selected for this purpose should be as similar as possible to what is encountered in practice. Based on this fundamental principle, we have distilled the following characteristics that we believe should be pursued when selecting or designing a dataset:

- **Accurate labels.** Label accuracy should take precedence over all other desirable benchmark qualities because inaccurate labels cast doubt on all derived results. Since anomalies are subjective by definition, dubious labelling is commonplace in the field and, to an extent, somewhat unavoidable. Crucially, we contend that it should be possible to *find some evidence* for all labelled anomalies within the time series data itself. After all, the goal should be to evaluate the algorithm's ability to process the given data, without access (or coincidental similarity) to external knowledge. This should go without saying, yet some benchmark problems (especially those derived from real-world phenomena) expect models to detect anomalies based on data which is loosely-related to the underlying system at best. Ideally, labels should also be externally validated by leveraging out-of-band information wherever possible. This is why we strongly emphasise the need for each anomaly to be substantiated in some way. Ideally, documentation should accompany every anomaly detection problem, with an explanation of how each anomaly differs from the normal data and how the labels are justified in terms of out-of-band information (where applicable).

- **A low anomaly density.** Anomalies are relatively rare, by definition, and benchmarking datasets should reflect that. If there is nearly as much "anomalous data" as normal data, then the definition of an anomaly is stretched too far, and perhaps classification methods would be more applicable. Although there is no fixed upper limit on the acceptable percentage of anomalous points, we believe that anything above 10% is excessive.

- **Non-trivial problems.** Wu & Keogh (2023) raised concerns about the triviality of many datasets in the literature. Although we do believe that simple problems have their place in datasets as a yardstick of baseline competency, the lack of more challenging problems in many datasets makes it difficult to discern the true capability of a model—clearly, anomaly detection tasks should cover the full gamut of difficulty. Achieving this balance is not simple due to the lack of a principled approach towards quantifying the difficulty of a given problem. Wu & Keogh (2023) proffered an intuitive definition based on whether a MATLAB "one-liner" using primitive functions is able to solve the problem. As the authors acknowledged themselves, this definition is far from perfect, since virtually any anomaly may be identified using so-called "magic numbers," as demonstrated in Appendix B. As a result, we contend that the usefulness of this definition is limited beyond illustrative examples, but do believe that this line of research (quantifying anomaly difficulty) would be very insightful.

- **Diverse anomaly types and locations.** Many different taxonomies of anomaly types have been proffered in the literature (Schmidl et al., 2022; Zamanzadeh Darban et al., 2024). The more diverse the anomalies, the lower the likelihood that a potential blindspot in the detection algorithm goes unnoticed. Similarly, anomalies should vary in location as well. Wu & Keogh (2023) noticed a *run-to-failure bias* in some datasets, where anomalous data is concentrated near the tail of the time series. This may result in a misleading evaluation of performance since location biases are not applicable to real-world streaming scenarios.

- **A large dataset size.** Larger datasets are more desirable since they tend to be statistically more significant. Naturally, small datasets are not useless, but they would need to be complemented by other datasets.

- **Realism.** Ideally, all other things being equal, benchmark problems sourced from real-world data are preferred to synthetic datasets.

The above criteria are, in some ways, a response to the flaws noted by Wu & Keogh (2023)—namely, mislabelling, triviality, unrealistic anomaly density, and run-to-failure biases. These criteria should be extended with additional emphases on anomaly diversity, dataset size, and realism in order to form a comprehensive list of desirable dataset characteristics.

## 5.2 Publicly available datasets

Based on the desirable characteristics outlined above, we now provide a brief description and critical review of the most prominent anomaly detection datasets in the literature. Our research here is largely inspired by, and an extension of, the critical analysis performed by Keogh (2021). We focus on univariate benchmark datasets here since these are easier to validate and illustrate our main point clearly: most time series anomaly detection datasets in the literature are unsuitable for benchmarking purposes, largely due to dubious labelling and high anomaly density.

A handful of archives (or dataset collections) have been made available in the time series anomaly detection literature. The TimeEval archive (Schmidl et al., 2022) is one such archive that comprises both univariate and multivariate datasets. The archive contains a new synthetic dataset called GutenTAG (Wenig et al., 2022), is well-structured and accompanied by detailed metadata, and curates a handful of public datasets as well. Regrettably, many of these public datasets are flawed due to problems including mislabelling, anomaly density, and overly trivial problems, as will be discussed later.

The TSB-UAD archive was introduced by Paparrizos et al. (2022b) and we employ their version of the respective datasets wherever available—their curation, collection, and standardisation of univariate datasets

proved to be invaluable. The archive comprises 13 766 individual time series problems, which are categorised into *public* (datasets previously made available in the literature), *artificial* (classification datasets which have been transformed into anomaly detection problems), and *synthetic* (augmented versions of the public datasets) subsets. We instead partition the reviewed datasets into three categories: those we recommend for use, those we only partially recommend (with some caveats), and those we do not recommend. These recommendations are supported by illustrative examples from each dataset in Appendix C.

The TSB-AD archive (Liu & Paparrizos, 2024) was recently released and may be seen as a successor to the TSB-UAD archive. It includes multivariate time series problems as well, while featuring many of the same univariate datasets as in its predecessor archive. The datasets in the archive have been subjected to additional automated and manual curation, resulting in an arguably more refined set of benchmark problems than those in the original TSB-UAD archive. Although the TSB-AD curation process represents a significant and commendable effort to improve the quality of datasets, we contend that some key limitations remain. The manual curation step is invaluable, but it is unfortunately not possible to validate since (to the best of our knowledge) the annotators' reasoning was not made available. Moreover, the curation still depends on the initial labels (some of which, without better documentation from the dataset creator, may be flawed beyond repair as suggested by Wu & Keogh (2023)). Although such large-scale curation efforts are invaluable, questions regarding label provenance and potential algorithm-specific bias remain.

In our subsequent review of the benchmarking landscape, we first examine datasets that should be avoided when benchmarking time series anomaly detection models. Mislabelling issues, high anomaly density, or excessive triviality may result in a skewed view of an anomaly detection model's performance. Some of the datasets may be repaired if a concerted effort is made, but in most cases this is not possible or worthwhile. We believe the following datasets should be avoided:

- **Dodgers loop** (Ihler et al., 2006). The single time series in this dataset representing traffic data suffers from questionable and highly subjective labels. This is because the labelled anomalies are expected bursts of traffic when a game finishes at the Dodgers Stadium. The Stadium also hosts rock concerts and other non-baseball sporting events that would have similar traffic dynamics, however, yet these are not labelled, resulting in many false negatives in the ground truth. Moreover, the labelled times seem to be nominal durations of the games, but if a game is heavily lopsided, many fans leave early, affecting the accuracy of the labels.

- **Numenta** (Lavin & Ahmad, 2015). This is a large and well-known benchmark with data sourced from diverse domains, but many time series are unfortunately plagued by mislabelling or triviality.

- **Sensorscope** (Yao et al., 2010). The time series in this dataset are drawn from a wide variety of environmental data such as temperature, humidity, and solar radiation, but unfortunately many of the labels are highly dubious.

- **The TSB-UAD artificial datasets** (Paparrizos et al., 2022b). These datasets were developed from existing classification datasets, but the conversion process resulted in various time series with highly dubious labels.

- **The TSB-UAD synthetic datasets** (Paparrizos et al., 2022b). This archive is based on anomaly detection problems derived from the corresponding public datasets which have been augmented with new or more difficult anomalies, but unfortunately the flaws in the public datasets are inherited and remain unresolved.

- **NASA SMAP** (Hundman et al., 2018). Time series in this dataset are sourced from the *Soil Moisture Active Passive* (SMAP) satellite and suffer from dubious labels, high anomaly density, and triviality.

- **NASA MSL** (Hundman et al., 2018). As in the case of NASA SMAP, this Mars Rover dataset features too many instances of questionable or overly trivial labels.

- **Daphnet** (Bächlin et al., 2009). The conversion process from the original multivariate data sourced from sensors attached to Parkinson's disease patients seem to have resulted in problems that are virtually unsolvable.

- **GHL** (Filonov et al., 2016). These univariate problems were converted from original multivariate data based on a simulated gasoil heating loop, resulting in anomalies that are very difficult to substantiate.

- **Genesis** (von Birgelen & Niggemann, 2018). This is a small univariate dataset which has been converted from an original, multivariate dataset monitoring a portable pick-and-place robot that also suffers from mislabelling.

- **OPPORTUNITY** (Roggen et al., 2010). The conversion from the original multivariate activity classification dataset resulted in many dubious labels.

- **Occupancy** (Candanedo & Feldheim, 2016). All the instances in this dataset (converted from an original multivariate classification dataset in which the task was to identify whether a room was occupied based on readings such as temperature, humidity, and light sensors) have unreasonably high anomaly density and some dubious labels as well.

- **SMD** (Su et al., 2019). This dataset represents metrics from server machines (such as CPU load, network usage, etc.) and was converted from an original multivariate dataset. It suffers from questionable labels and, in some cases, triviality.

- **ECG** (Paparrizos et al., 2022b). The *electrocardiogram* dataset was created by the authors of the TSB-UAD archive (Paparrizos et al., 2022b) by partitioning one long ECG time series from the MIT-BIH dataset. This dataset is fundamentally flawed for benchmarking because the anomalies present are largely repetitions of the same specific arrhythmia. As a result, a model need only identify a single instance to detect trivially the remainder via similarity matching. High scores on this dataset reflect the abundance of these repeated events rather than the model's ability to detect true anomalies, leading to a massive overestimation of performance.

- **MITDB** (Moody & Mark, 2001). This collection of 48 half-hour ECG recordings is widely used but suffers from the same critical weakness as the dataset above. Many records contain thousands of instances of the exact same anomaly type. Standard evaluation metrics on this dataset inherently "overcount" success, rewarding an algorithm many times for learning a single, often simple, pattern. This redundancy obscures whether a model can actually generalise to diverse or subtle anomalies, rendering the dataset unsuitable for rigorous benchmarking.

Even flawed datasets which violate some of the desirable qualities outlined in §5.1 might have some use. The datasets listed below should be suitable for benchmarking purposes, with some caveats and after having addressed some of the more serious concerns:

- **IOPS** (Paparrizos et al., 2022b). As preprocessed in the TSB-UAD archive, this dataset consists of 29 anomaly detection problems together with labelled training data for each one. The reasoning behind the anomalies is not documented to the best of our knowledge, but a visual inspection of the problems indicates that the labels appear to be reasonable. Some problems are trivial, but this does not discount the entire dataset.

- **Yahoo** (Laptev et al., 2015). The A1 subset of the Yahoo dataset contains real data and should be retained, while the A2, A3, and A4 subsets represent anomaly detection problems that are too simple. A few instances of mislabelling in the A1 subset can be fixed (simply by removing A1–32, A1–35, A1–46, A1–47, and A1–67), and the run-to-failure bias is not that severe.

- **SVDB** (Greenwald et al., 1992). The Supraventricular Arrhythmia Database also appears to be a good benchmark based on real-world cardiological data, after problems with a high anomaly density are filtered out.

Our list of recommended datasets is short. Since it is our contention that the single, most important requisite quality for a good time series anomaly detection benchmark is accurate anomaly labels, it is no surprise that the labelling processes in the recommended datasets are well-substantiated:

- **The UCR time series anomaly archive** (Keogh, 2021; Wu & Keogh, 2023) is arguably the best univariate dataset available at present. Each instance is documented, and each anomaly is justified. Moreover, the anomalies are diverse in terms of both type and location, while also being non-trivial for the most part. Each problem only contains one anomaly by design so that anomaly scoring models can be evaluated in isolation using the UCR score defined in §4.1. This, however, is also a drawback, since researchers or users who wish to evaluate and compare thresholding approaches need to look elsewhere for datasets with multiple anomalies in each time series.

- **The Mackey Glass benchmark** (Thill et al., 2020) addresses the need for high-quality datasets with multiple anomalies in each time series. Although the data and the anomalies here are synthetic, the generation methodology underlying the dataset is well-documented and sensible. While the dataset is relatively short, a tool is made available so that users may create similar anomaly detection problems.

- **The synthetic GutenTAG dataset** (Schmidl et al., 2022; Wenig et al., 2022) was introduced as part of the TimeEval archive. It consists of 193 datasets in total, with a mix of univariate and multivariate time series. The datasets were created by a set of base oscillations aimed at generating "normal data," before perturbing the time series using a set of anomaly injectors. The problems are relatively easy, but otherwise it is a high quality benchmark.

The key metadata for the three recommended datasets are summarised in Table 1. The low anomaly sequence count and overall contamination level align with the fundamental requirement that anomalies remain rare occurrences within benchmark datasets.

Table 1: Metadata for the recommended datasets

|  | GutenTAG | MackeyGlass | UCR |
|---|---|---|---|
| Problem count | 168 | 10 | 250 |
| Anomalous sequence count | 442 | 100 | 250 |
| Data realism | Synthetic | Synthetic | Real |
| Anomaly realism | Synthetic | Synthetic | Synthetic |
| Average train length | 10 000 | 257 | 21 209.8 |
| Average test length | 10 000 | 99 743 | 56 205.3 |
| Average contamination | 3.5% | 4% | 2.4% |
| Average anomalous sequences | 2.5 | 10 | 1 |

More detailed quantitative profiling for the remaining datasets may be found in the TSB-AD archive (Liu & Paparrizos, 2024).

### 5.3 Dataset suggestions

A thorough examination of prominent time series anomaly detection datasets revealed that most suffer from some form of mislabelling. Some of these dubious labels were introduced *via* the TSB-UAD conversion process from multivariate data, which involves turning each dimension into a separate anomaly detection problem, and then filtering out problems in which none of the algorithms tested was able to achieve an $AUC_{ROC}$ score above 0.8. We have two reservations about this approach. Apart from our previously substantiated view (described in §4.1) that $AUC_{ROC}$ is a poor metric for this task, we also believe that filtering anomaly detection problems based on model performance could be seen as an example of circular reasoning, since these very same models are then meant to be benchmarked on the dataset later. A model's output cannot

ascertain the accuracy of the ground truth—once again, it would be better if some documentation, reasoning, or substantiation behind the labels was provided by the dataset creators.

Let us finally turn our attention to how datasets should be developed in practice. We believe there are three broad approaches that may be adopted:

- **Real data and anomalies.** High-quality datasets sourced from actual phenomena are ideal because of their inherent realism. Due to the time-consuming and difficult nature of curating labels, as well as the often expensive underlying data-generating system, this approach tends to be costly. The high cost, as well as the potentially sensitive nature of the data, means that few such datasets are made available publicly, with relatively rare exceptions such as the NASA datasets (Hundman et al., 2018). Moreover, one has to be particularly careful when preprocessing real data so that the qualities outlined in §5.1 are met.

- **Real, normal data with synthetic anomalies added.** This approach is simpler and more cost-effective since it is typically easier to ensure that data is free of anomalies than to identify their locations. Once normal data is available, anomalies may be introduced by perturbing the time series in some way. When executed well, as in the case of the UCR archive (Keogh, 2021; Wu & Keogh, 2023), this approach successfully balances realism with the other desirable qualities in §5.1.

- **Synthetic data and anomalies.** This paradigm involves injecting artificial anomalies into synthetic base oscillations which are considered the normal data. This is the quickest, easiest, and cheapest approach, while providing the most control over the anomaly detection problems generated (e.g. one may modify the anomaly types, lengths, locations, and density to one's liking). This comes at the expense of realism—since everything is synthetic, there is little that tethers each problem to real-world applications. Another challenge is ensuring that there are at least a few anomaly detection problems in the dataset which are truly difficult, since many synthetic archives are overly simple. An intuitive method of enhancing difficulty involves adding noise to base oscillations (as suggested in the Mackey Glass datasets (Thill et al., 2020)), but we believe that this approach is conceptually risky since a corrupted data-generating process means that the accuracy of labels can no longer be guaranteed—it can become difficult to verify that a supposed anomaly is not merely a byproduct of injected noise. This is not meant as a critique of the Mackey Glass datasets (in which the amount of injected noise is relatively low), but rather as a precautionary guideline. Instead, we suggest adjusting the difficulty of anomaly detection problems by generating arbitrarily complex base oscillations (for instance, by exploiting differential equations that exhibit chaotic behaviour in their solutions) and increasing the subtlety of injected anomalies thereafter. Since any deviation from a deterministic base oscillation may be considered an anomaly by definition, these anomalies may be as subtle as desired without compromising the integrity of the labels.

Regardless of the approach selected, we strongly emphasise the need for future datasets to be well-documented and anomalies to be substantiated in some way. This is essential in instilling trust in the accuracy of the labels.

## 6 Model alignment

Most research in time series anomaly detection is focused on developing novel detection models and methods. The latest models are often complicated, involve some flavour of deep learning, and require careful tuning of their hyperparameters. A growing body of research, however, indicates that these complex models are often overkill, and perform poorly once re-evaluated in respect of better datasets with more appropriate metrics (Rewicki et al., 2023; Wu & Keogh, 2023).

These complex models are often also not compared with sensible baselines. Sarfraz et al. (2024) proposed a few simple and effective baselines, including sensor range deviation, nearest neighbour distance (to the nearest training data), the *principal component analysis* (PCA) reconstruction error, and four neural network blocks which represent prominent deep learning architectures.

Algorithms based on time series discords and the matrix profile (Yeh et al., 2016; Lu et al., 2023), as well as the simple baselines mentioned above, often outperform far more complex approaches. The success of these baselines begs the question whether there is a true need for more powerful models—in other words, perhaps most time series anomaly detection problems in practice may be solved by simple means. Consider many of the applications outlined in §3.1, such as monitoring information technology systems or a patient's vital signs—typical anomalies would take the form of large spikes or drops in the sensor values, which may be detected by basic methods. It may, of course, be argued that more powerful algorithms will unlock new potential applications, but it should be kept in mind that existing algorithms are often accurate enough for many current use cases.

As a result, we suggest that a greater research focus be placed on improving computational efficiency (ensuring that models are fast enough for real-time detection), model interpretability, and explainability of the predictions. It is important to instill trust in the system by keeping the user as informed as possible. Much of the research in this area is limited to univariate data (Jacob et al., 2021; Der et al., 2024), so multivariate extensions, as well as improved techniques in general, would be particularly useful in practice.

## 7 Conclusion and suggested research avenues

Much of the research in time series anomaly detection suffers from unsuitable performance metrics, flawed datasets, and poor benchmarking practices. In each case, we examined the literature and provided suggestions for future research directions in order to align the theory with its practical applications in domains such as healthcare, predictive maintenance, finance, and cybersecurity.

We showed that many popular metrics, such as the point-wise $F_1$ score, the point-adjusted $F_1$ score, the $AUC_{ROC}$, and the $AUC_{PR}$, exhibit disqualifying flaws. Instead, we proposed the realistic $F_1$ score and realistic $AUC_{PR}$, which reward correctly identified anomalous events once, penalise each false positive, and ignore redundant true positives for anomalies which have already been detected.

Thereafter, we critiqued some of the most prominent datasets, and showed that many are plagued with labelling problems. In order to help remedy this predicament, we identified desirable characteristics in datasets, including accurate labels, low anomaly densities, non-trivial anomaly detection problems, realism, and diverse anomaly types and locations. We also provided practical suggestions pertaining to the development of new datasets.

Many newly proposed algorithms are often evaluated by adopting poor benchmarking practices, and without being compared with simple baselines. Much research effort has been devoted to developing more powerful models, but we contend that existing models are often accurate enough already for most practical applications. Instead, we recommended a renewed effort to improve models in terms of computational efficiency, interpretability, and explainability so that they become more useful in practice.

A relatively unexplored research direction is improving our understanding of what makes certain anomaly detection problems difficult. This would help identify the relative strengths and weaknesses of models, and aid in developing better datasets which span the full spectrum of difficulty.

Beyond academic rigour, the choice of evaluation methodology carries significant implications for safety-critical deployments. In domains such as healthcare monitoring and industrial safety, the reliance on inflated metrics (such as the point-adjusted F-score) creates a dangerous false sense of security, potentially allowing models to be deployed that fail to detect the onset of catastrophic events despite reporting high benchmark scores. By shifting towards realistic evaluation frameworks, the community can ensure that reported performance aligns with operational reality, ultimately leading to more reliable and trustworthy systems in high-stakes environments.

Although we hope that our suggestions will help steer research in a direction that is more aligned with practical applications, we recognise the limitation of our work, and welcome challenges and critique. Our primary goal is merely to call attention to building blocks of the field that have been neglected, and hopefully spark debate as to what the best practices in time series anomaly detection truly are.

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

# A Comparing evaluation metrics

To substantiate the critical analysis presented in §4, we conducted a systematic experiment designed to isolate the failure modes of standard evaluation metrics and demonstrate the practical utility of the proposed alternatives. Specifically, we benchmarked the point-wise F-score ($F_{1,p}$), event-wise F-score ($F_{1,e}$), point-adjusted F-score ($F_{1,a}$), composite F-score ($F_{1,c}$), and the realistic F-score ($F_{1,r}$) across the three datasets recommended in §5: the UCR anomaly archive, the Mackey-Glass benchmark, and the GutenTAG synthetic dataset.

We evaluated six distinct anomaly scoring approaches, selected to represent a diverse range of methodologies. These include the local outlier factor (LOF) (using $k = 50$ neighbours and window size $w = 50$), matrix profile (MP), autoencoder, and the fast Fourier transform, as well as two naive baselines: a *random scorer* (to test for noise rejection) and a *flag all* baseline (to test for event-precision).

All algorithms (except for the two naive baselines) were implemented using the TSB-AD library (Liu & Paparrizos, 2024) with default hyperparameters. To ensure a fair comparison that reflects the "best-case" potential of each model, we performed a grid search over anomaly thresholds $u \in \{0.05, 0.1, \ldots, 0.95\}$. Furthermore, we evaluated each configuration both with and without the postprocessing step described by Barrish & van Vuuren (2026) (which filters predictions to retain only the first threshold crossing or local peaks in anomaly scores). The results presented in Table 2 reflect the average maximum F-score achieved by each algorithm's optimal configuration on each dataset, with the rank of each anomaly scorer shown in parentheses. This protocol ensures that low scores are attributable to the fundamental misalignment of the metric or the model's output, rather than suboptimal thresholding.

The results presented in Table 2 substantiate the concerns regarding metric reliability raised in §4. Most notably, the point-adjusted metric ($F_{1,a}$) assigns a purely random scoring function an F-score of 0.89 on the Mackey-Glass dataset. This confirms that point-adjustment renders the metric incapable of distinguishing between state-of-the-art detection and random noise, as a random distribution will eventually hit a portion of every anomaly and trigger full credit. In contrast, the realistic F-score ($F_{1,r}$) correctly identifies this behaviour as noise, assigning it a score of 0.02.

Similarly, the event-wise metric ($F_{1,e}$) proves susceptible to trivial gaming. The *Flag All* baseline achieves a perfect score of 1.00 across the UCR and Mackey-Glass datasets under $F_{1,e}$ simply by flagging every time step. Because the $F_{1,e}$ metric fails to penalise false positive duration, it disproportionately rewards such "alarmist" behaviour. The recommended $F_{1,r}$ accounts for this precision failure, correctly assigning the baseline a score of 0.00.

Crucially, the choice of metric fundamentally alters the leaderboard. Under the inflated $F_{1,a}$, the autoencoder appears highly competitive on the UCR archive (0.84), nearly matching the matrix profile (0.94). Under realistic evaluation ($F_{1,r}$), however, the autoencoder's performance collapses to 0.30, while the matrix profile remains robust at 0.64. This suggests that the autoencoder's success under standard metrics is largely driven by "fragmented", sporadic alerts.

Finally, the results illuminate the fundamental philosophical distinctions between point-wise, composite, and realistic evaluation. Standard point-wise evaluation ($F_{1,p}$) treats anomaly detection as a binary classification task at every timestamp. While theoretically rigorous, this approach is operationally brittle: it penalises valid detections for minor latency or misalignment, often assigning low scores to models that successfully flagged the event.

Table 2: Average F-scores and ranks (in parentheses) across three datasets

| Dataset | Algorithm | $F_{1,p}$ | $F_{1,e}$ | $F_{1,a}$ | $F_{1,c}$ | $F_{1,r}$ |
|---------|-----------|-----------|-----------|-----------|-----------|-----------|
| GutenTAG | LOF | 0.71 (**1**) | 0.83 (**2**) | 0.94 (**1**) | 0.84 (**1**) | 0.75 (**1**) |
| GutenTAG | MP | 0.51 (**2**) | 0.76 (**4**) | 0.88 (**2**) | 0.65 (**2**) | 0.55 (**2**) |
| GutenTAG | FFT | 0.28 (**4**) | 0.57 (**5**) | 0.71 (**4**) | 0.49 (**4**) | 0.41 (**3**) |
| GutenTAG | AE | 0.33 (**3**) | 0.80 (**3**) | 0.79 (**3**) | 0.52 (**3**) | 0.39 (**4**) |
| GutenTAG | Random | 0.06 (**5**) | 0.05 (**6**) | 0.70 (**5**) | 0.07 (**5**) | 0.03 (**5**) |
| GutenTAG | Flag all | 0.05 (**6**) | 1.00 (**1**) | 0.05 (**6**) | 0.06 (**6**) | 0.00 (**6**) |
| MackeyGlass | LOF | 0.14 (**1**) | 0.91 (**2**) | 0.96 (**1**) | 0.93 (**1**) | 0.91 (**1**) |
| MackeyGlass | MP | 0.13 (**2**) | 0.75 (**4**) | 0.91 (**2**) | 0.77 (**2**) | 0.74 (**2**) |
| MackeyGlass | AE | 0.09 (**3**) | 0.80 (**3**) | 0.79 (**5**) | 0.18 (**3**) | 0.13 (**3**) |
| MackeyGlass | FFT | 0.08 (**4**) | 0.11 (**5**) | 0.85 (**4**) | 0.13 (**4**) | 0.08 (**4**) |
| MackeyGlass | Random | 0.08 (**5**) | 0.02 (**6**) | 0.89 (**3**) | 0.09 (**5**) | 0.02 (**5**) |
| MackeyGlass | Flag all | 0.08 (**6**) | 1.00 (**1**) | 0.08 (**6**) | 0.08 (**6**) | 0.00 (**6**) |
| UCR | LOF | 0.29 (**2**) | 0.71 (**3**) | 0.93 (**2**) | 0.74 (**1**) | 0.67 (**1**) |
| UCR | MP | 0.40 (**1**) | 0.72 (**2**) | 0.94 (**1**) | 0.71 (**2**) | 0.64 (**2**) |
| UCR | AE | 0.13 (**3**) | 0.58 (**4**) | 0.84 (**3**) | 0.37 (**3**) | 0.30 (**3**) |
| UCR | FFT | 0.10 (**4**) | 0.27 (**5**) | 0.84 (**4**) | 0.30 (**4**) | 0.23 (**4**) |
| UCR | Random | 0.05 (**5**) | 0.02 (**6**) | 0.70 (**5**) | 0.06 (**5**) | 0.02 (**5**) |
| UCR | Flag all | 0.05 (**6**) | 1.00 (**1**) | 0.05 (**6**) | 0.05 (**6**) | 0.00 (**6**) |

The composite F-score ($F_{1,c}$) attempts to bridge this gap by combining point-wise precision with event-wise recall. It still rewards flagging long anomalies multiple times, however, which is arguably undesirable in most practical applications. Instead, realistic F-score ($F_{1,r}$) represents a paradigm shift towards operational utility. It is built on the premise that an operator needs to be alerted to an event once. By treating the event as the fundamental unit of recall (like $F_{1,e}$) but strictly penalising the false positive rate of the alert duration (unlike $F_{1,e}$), $F_{1,r}$ effectively filters out "spammy" models.

## B  Defining the notion of anomaly detection "triviality"

Wu & Keogh (2023) were not only amongst the first researchers to note the rampant problem of overly trivial benchmarks, but they also took this a step further by proffering a pragmatic definition of a *trivial anomaly detection problem*:

**Definition B.1** (Trivial time series anomaly detection problems). "A time series anomaly detection problem is *trivial* if it can be solved with a single line of standard library MATLAB code. We cannot 'cheat' by calling a high-level built-in function such as *kmeans* or *ClassificationKNN* or calling custom written functions. We must limit ourselves to basic vectorized primitive operations, such as *mean, max, std, diff, etc.*"

While this definition serves as an excellent heuristic for identifying obviously simple problems, its strict application presents operational challenges, as the authors acknowledged themselves. A key nuance lies in distinguishing between generalisable domain heuristics and *post-hoc* parameter tuning. Without this distinction, reliance on specific "magic numbers" allows one to offer a one-liner justification for *any* time series anomaly, provided the anomaly's location is known *a priori*.

As an example, it is worth examining the UCR time series anomaly archive (arguably the best time series anomaly detection benchmark dataset), introduced by Wu & Keogh (2023). This truly is a difficult dataset—few models tested on the dataset in the literature are able to solve more than 50% of the problems correctly. Within this dataset, problem 001 was shown to be particularly difficult for many models. This anomaly depicted in Figure 1, however, yields to a "simple" one-liner that satisfies Definition B.1 (the line of code is admittedly in Python, but it could easily be translated to MATLAB):

```
np.where(np.round(np.diff(values), 6) == 8.69513, 1, 0)
```

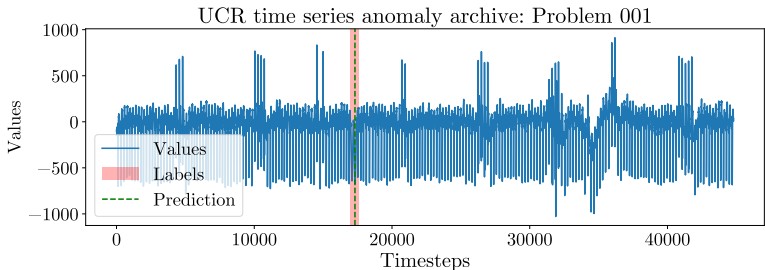

Figure 1: An example of a difficult anomaly detection problem, solved by a "simple" one-liner.

Of course, the one-liner above is exceptionally contrived, since the 8.69513 constant is chosen based on prior knowledge of where the anomaly is. One could solve this problem by employing various other one-liners as well, but the purpose of this example is not to classify the problem as trivial. Rather, it highlights the difficulty in defining the mathematical boundary between a valid simple solver and one that "overfits" via magic numbers. Consequently, while Definition B.1 is powerful for filtering out noise, identifying these illustrative one-liners relies on human judgement to determine whether a solution represents a legitimate heuristic or a contrived fit.

The shortcomings of Definition B.1 do, however, highlight the need for some principled method for quantifying the difficulty of time series anomaly detection problems. Given such a difficulty measure, one could empirically ensure that datasets possess anomaly detection problems that span a wide range of difficulty. Moreover, one could compare entire datasets in terms of their overall difficulty. This line of research would clearly be very useful, and has, to the best of our knowledge, not been thoroughly examined in the literature.

## C  Gallery of illustrative benchmark problems from public datasets

In this appendix, we present illustrative examples from the public datasets cited in this paper in order to support our views and criticisms of each.

### C.1  Datasets to avoid

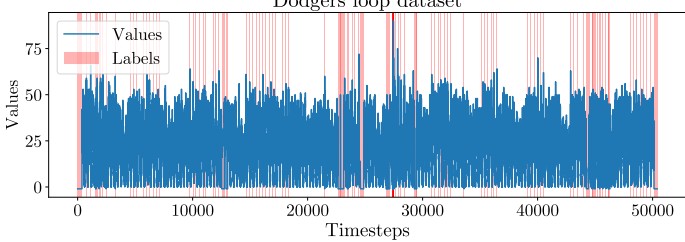

Figure 2: The entirety of the Dodgers loop benchmark dataset. Anomalies in the traffic data caused by Dodgers games are largely indistinguishable from other causes, such as car accidents.

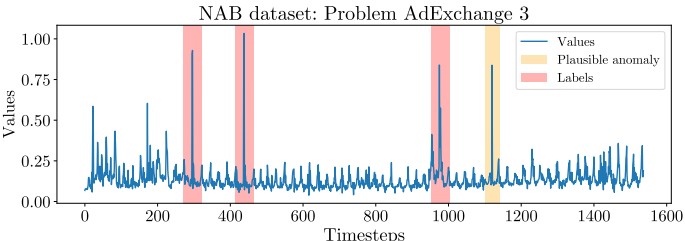

Figure 3: Problem 3 in the AdExchange section of the Numenta benchmark. The accuracy of the labels is somewhat questionable, since one of the largest spikes in the time series is not flagged.

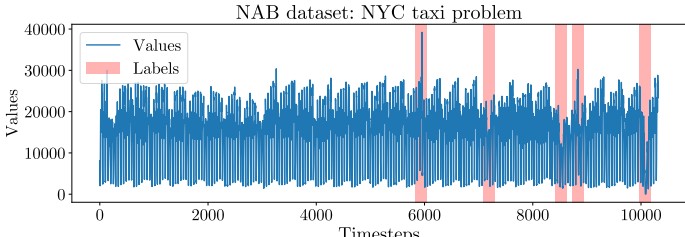

Figure 4: The NYC taxi problem in the Numenta dataset. Five anomalies are labelled, corresponding to the NYC marathon, Thanksgiving, Christmas, New Year's day, and a snow storm, but these may be seen as highly subjective since other anomalous events occurred during this period, as shown by Wu & Keogh (2023).

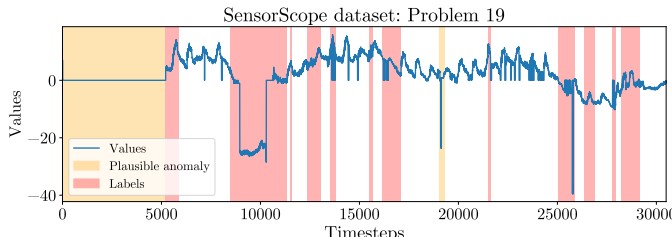

Figure 5: Problem 19 in the SensorScope dataset. Many of the anomalous labels are questionable, while a few normal subsequences (such as the long flatline period at the beginning and a sharp drop later) subjectively look as if they could pass as anomalies.

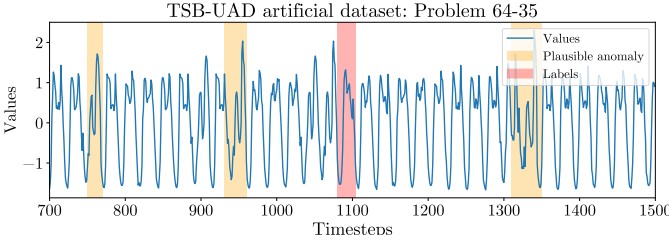

Figure 6: A snippet from a sample problem in the TSB-UAD artificial dataset. Numerous subsequences appear to be more anomalous than the labelled anomaly.

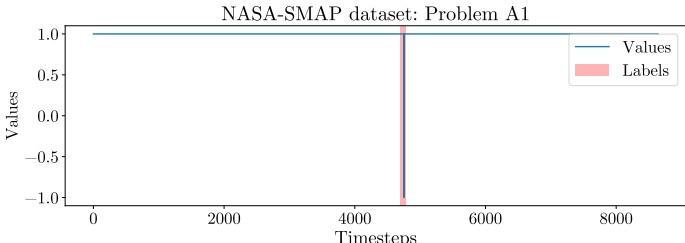

Figure 7: The A1 problem in the NASA SMAP dataset. It features a single, exceedingly simple anomaly.

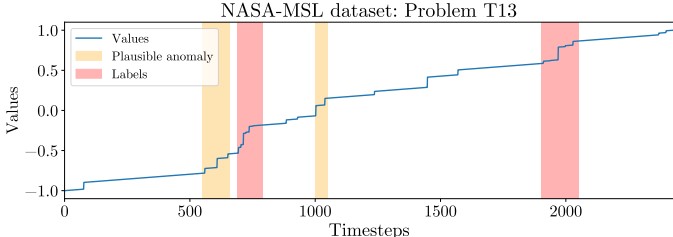

Figure 8: An example of a time series anomaly detection problem in the NASA MSL dataset with two dubious anomaly labels.

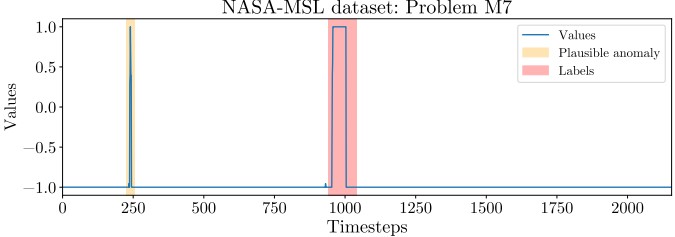

Figure 9: The M7 problem in the NASA MSL dataset. Two large spikes are visible in the data, but only one of these is labelled as an anomaly.

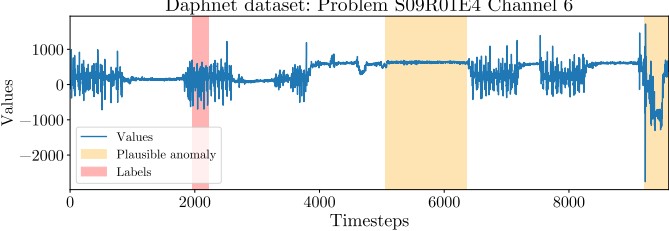

Figure 10: A sample problem from the Daphnet benchmark dataset. The original labelled data might be accurate, but it seems exceptionally difficult to motivate that this anomaly can be identified without other dimensions or external information. The long flatline subsequence or the large drop near the end of the time series could potentially be considered anomalies instead.

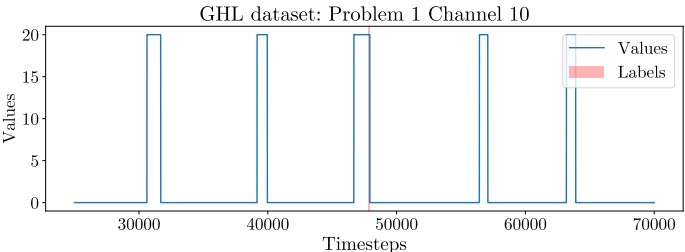

Figure 11: A sample anomaly detection problem from the GHL benchmark dataset. It is virtually impossible to justify the labelled anomaly using this univariate data alone.

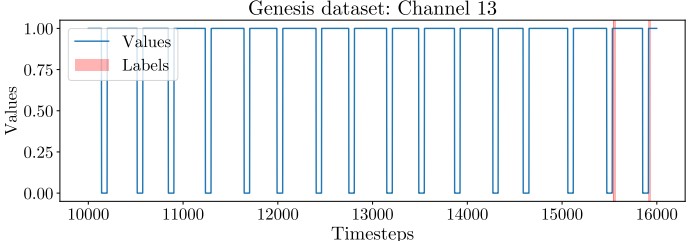

Figure 12: A sample anomaly detection problem in the Genesis benchmark dataset. It is virtually impossible to justify the labelled anomaly when considering the univariate data alone.

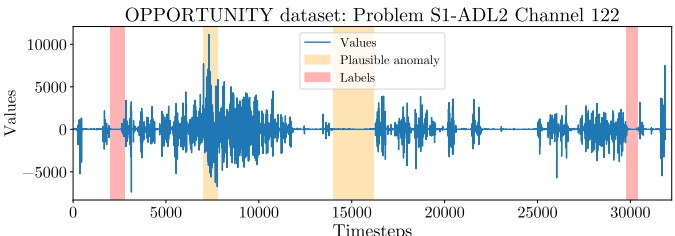

Figure 13: A sample anomaly detection problem in the OPPORTUNITY benchmark dataset. The transformation from a classification task to an anomaly detection task results in many instances of questionable labels. The large spike or long flatline period could potentially be considered anomalous.

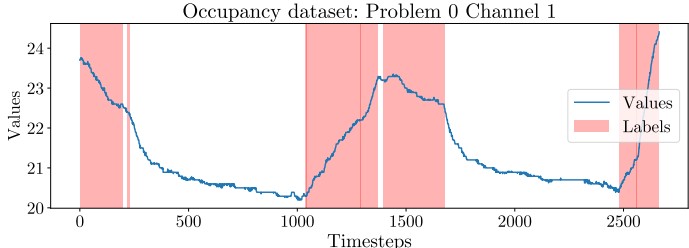

Figure 14: A sample anomaly detection problem in the Occupancy benchmark dataset. The anomaly density is unreasonably high, while the labels themselves are also of dubious accuracy.

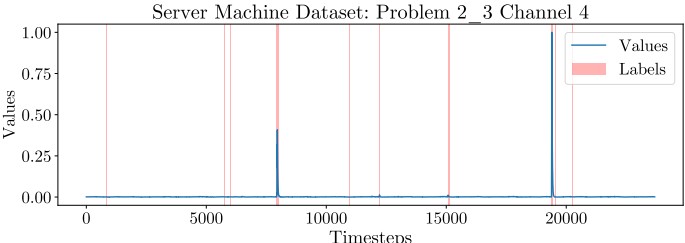

Figure 15: A sample anomaly detection problem in the SMD benchmark. Except for the two clear spikes, it is difficult to justify the other labelled anomalies.

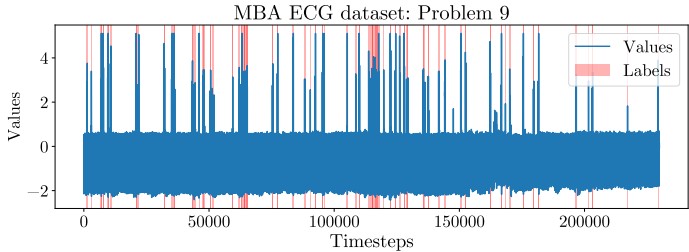

Figure 16: A sample problem in the ECG dataset. While the anomalies are distinct, they represent repetitions of the same arrhythmia. This allows models to achieve high scores via simple pattern matching rather than genuine anomaly detection.

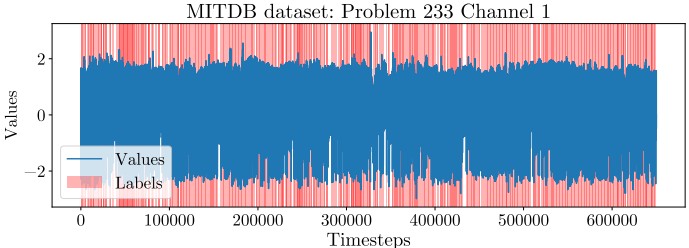

Figure 17: An example of an anomaly detection problem in the MITDB benchmark with a particularly high anomaly density. Although it is not quite as egregious as the figure makes it out to be, more than 34% of points are labelled as anomalies, which begins to stretch the definition of an "anomaly."

## C.2 Partially recommended datasets

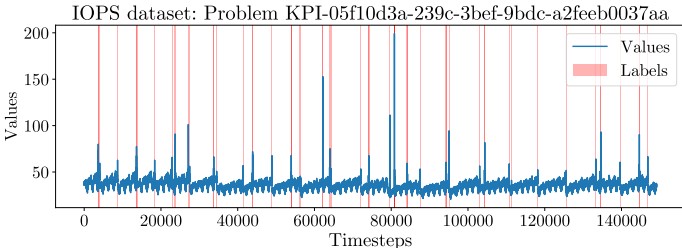

Figure 18: A sample anomaly detection problem in the IOPS dataset. The labels are relatively clear-cut.

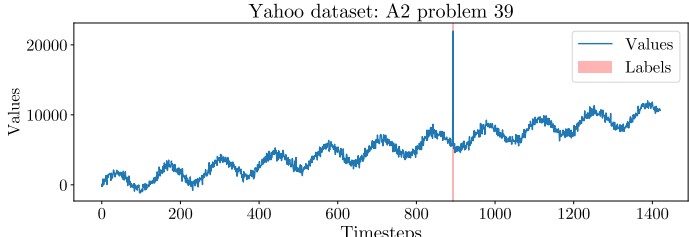

Figure 19: Problem 39 in the A2 subset of the Yahoo dataset. The single labelled anomaly takes the form of a large spike which is easy to identify.

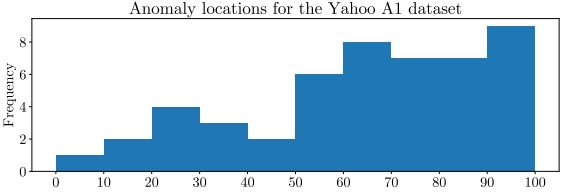

Figure 20: The anomaly locations within the Yahoo A1 subset as a percentage of the total time series length. The concentration of the anomaly labels near the tail end of the time series indicates that a run-to-failure bias exists. Adapted from Wu & Keogh (2023).

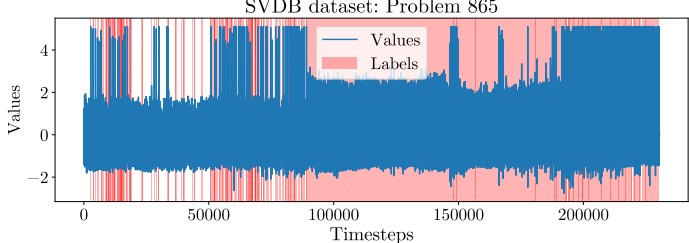

Figure 21: Problem 865 in the SVDB dataset. The dataset has an unreasonably high anomaly density of 60.5%.

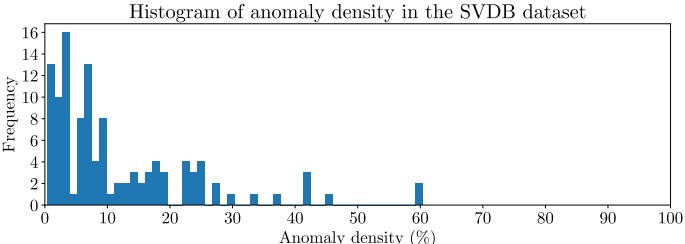

Figure 22: A histogram of the anomaly density of each problem in the SVDB dataset. It stands to reason that a decent portion of the dataset is still usable once those problems with high anomaly density have been removed.

### C.3 Recommended datasets

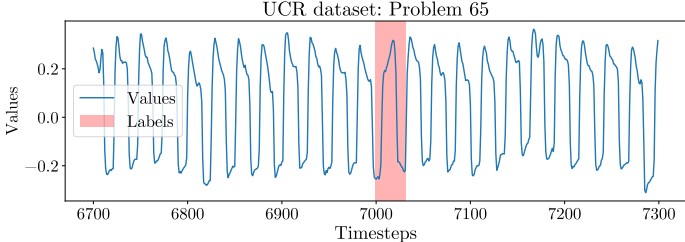

Figure 23: A sample anomaly detection problem in the UCR dataset. The anomaly is inserted by reversing a cycle in the original data. This is a good example of an anomaly which is well-substantiated, while also not overly simple for baseline models.

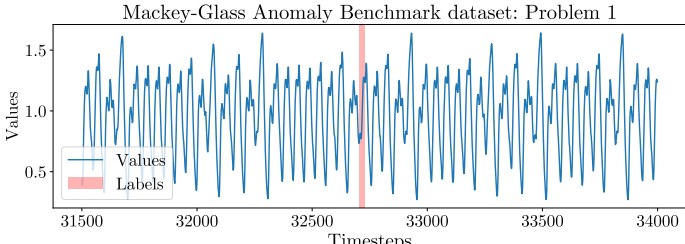

Figure 24: A sample anomaly detection problem in the Mackey Glass dataset. The anomaly is introduced by removing a subsequence from the original data and then stitching the time series back together, resulting in an anomaly which is very difficult for a human to identify.

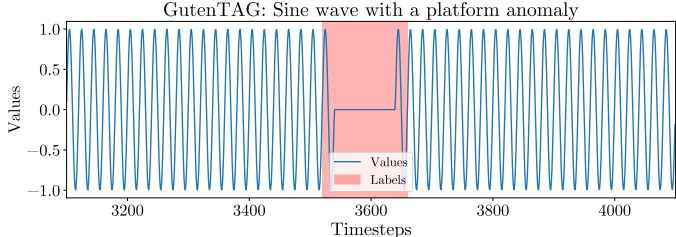

Figure 25: A sample anomaly detection problem in the GutenTAG dataset. The anomaly is introduced by introducing a "flatline" into the original sine wave. The dataset is well-documented and principled, but some problems can be trivial.

