# OpenReview forum: "Aligning time series anomaly detection research with practical applications"
_TMLR — Accepted by TMLR_

### Review · Reviewer_nbyJ · 2025-10-16

**Summary Of Contributions:**

The paper echoes and updates criticisms of TSAD by Wu and Keogh and others.

**Additional Comments:**

The paper largely restates Wu and Keogh (to be fair, it mostly admits that).

It is not clear what this paper adds beyond Wu and Keogh. For example, there are no new experiments or examples.

In spite of that, I weakly lean to accept, because it is clear that the community needs to hear the message, most of TSAD is rendered meaningless by poor metrics and datasets.

**Audience:**

Yes

**Audience Explanation:**

TSAD is still a hot topic, and that is not going to go away soon.

**Claims And Evidence:**

No

**Claims Explanation:**

The paper seems somewhat naive in not noticing that many problems it correctly summarizes still apply to The TSB-AD archive

**Requested Changes:**

“The former is concerned with determining a score for each timestep in the series, with 0 representing the most normal data and 1 representing the most anomalous”
Using a range of 0 to 1 is a good idea, but to be clear, this is not standard.

---
“Unfortunately, this metric has limited applicability since few benchmark datasets in practice have only one anomaly in each time series.”
This is not true, you misunderstand the point that was made.  You can trivially extend this scoring function to multiple anomalies.
Keogh’s two points about using this score are:

1)	Suppose you have (using text for a proxy):

…aaaaaaaaaaaaaaXaaaaaaaaaaaaaXaaaaaaaa….

And you find the first anomaly “X”. Should you also claim credit for finding the second anomaly “X”?

Many people do (see [a]) but that is clearly unfair, you are overcounting success, they are the same basic anomaly, if you find one, you will find the other. Almost all of the ECG (and some other) datasets of Paparrizos have this problem. Keoghs one anomaly per dataset complete bypasses this problem.


2)	The second point was that the success must be binary. Look at [b], it explains this perfectly. Most scoring functions assume precise duration of labels can be known, but that is almost never true.

Almost all the scoring functions you list suffer from the incorrect “precise duration of labels can be known”, and as such are worthless.

---

“Experts investigate each alert soon after the potential anomaly is flagged. Within a reasonable amount of time, these experts can determine whether a true anomaly was flagged based on their subjective interpretation of “usefulness.” One flag during an anomalous subsequence is sufficient to uncover the entire anomaly, and subsequent alerts are redundant.” There is a short video that illustrates this nicely [a]
I think you could modify your text to say : “One flag during an anomalous subsequence is sufficient to uncover the entire anomaly, and subsequent alerts are redundant. Moreover, if a paper does flag subsequent alerts, it is essentially overcounting success (see [a])”

---

“Although there is no fixed upper limit on the acceptable percentage of anomalous points, we believe that anything above 10% is excessive.”
I think 10% is two or three orders of magnitude too high. Look at [c], at the 15:0) timestamp. “One anomaly a year is not good, two…”

--

“Perhaps the main flaw is its reliance on so-called “magic numbers”—it is possible to offer a one-liner post-hoc justification for literally any time series anomaly”

I think this is a very unfair criticism and a real weakness of the paper.
There is a huge difference between the following “one-liners”
1)	diff(k170(:,2)) > 500                                                  (page 36 of [d])
2)	abs(diff(a)) >1                                                             (page 49 of [d])
3)	sensor(:,7) < 100                                                      (page 61 of [d])
And a one-liners like
A)	np.where(np.round(np.diff(values), 6) == 8.69513, 1, 0)


---

“The TSB-AD archive (Liu & Paparrizos, 2024) was recently released ..have been subjected to additional automated and manual curation, resulting in an arguably more refined set of benchmark problems than those in the original TSB-UAD archive”
Sorry, but that reads as hopelessly naïve. The TSB-AD archive is plagued with problems (and dubious provenance to boot).
“This is not meant as a critique of the stellar work that went into the invaluable TSB-AD archive,” Again, I don’t mean to harp on this, but it is not “stellar work”, there was almost no introspection, and the datasets appear to have been chosen to favor Paparrizos algorithms, not for any other reason.  It is not “invaluable”, it is a negative contribution that is holding the community back.

--


“Dodgers loop (Ihler et al., 2006). The single time series in this dataset representing traffic data suffers from questionable and highly subjective labels.”
This might be too vague to scare off users.  A better bit of text here would be:
 “Dodgers loop (Ihler et al., 2006). The single time series in this dataset representing traffic data suffers from questionable and highly subjective labels. This is because the “anomalies” are the expected bursts of traffic when a game finishes at the Dodgers Stadium. However, the Stadium also hosts rock concerts and other non-baseball sporting events that would have similar traffic dynamics, yet they are not labeled, resulting in many false negatives in the ground truth. Moreover, the labeled times seem to be nominal times/durations for games. But if a game is heavily lobsided, many fans leave early. A visual inspection shows that the alignment of labels with “bursts, it weak and inconsistent.”.
More generally, these section (all of page 8 and 9) have been nicely illustrated and explained in [d]. So this seems redundant.

--

“Our list of recommended datasets is short…. The synthetic GutenTAG dataset”
Hmm, I think the community should set its sights higher than the synthetic GutenTAG dataset. It is trivial.

---

“Synthetic data and anomalies. This paradigm involves injecting”
The problem with this is that the people making the Synthetic data and anomalies are typically doing it to test THEIR algorithm. So they will make data that suits their algorithm.
The only way “Synthetic data and anomalies” would work is if an international committee of researchers from different institutions (perhaps at a workshop) came together, and create the data as a team.

---

“Partially recommended datasets” “MBA ECG dataset”
This is a VERY bad dataset. People treat it like there are dozens of different anomalies, but they are all the same arrhythmia, if you can detect one, you can detect them all. See [a].
Same is true for MITDB dataset: Problem 233 Channel 1 and for almost all EEG datasets.

--

Fig 19 is just copied from Wu and Keogh.


[a] https://www.youtube.com/watch?v=fR4vqwmALZM
[b] https://www.youtube.com/watch?v=3gH-65RCBDs
[c] https://www.youtube.com/watch?v=Vg1p3DouX8w
[d] https://www.dropbox.com/scl/fi/cwduv5idkwx9ci328nfpy/Problems-with-Time-Series-Anomaly-Detection.pdf?rlkey=d9mnqw4tuayyjsplu0u1t7ugg&dl=0

---

> ### Author Response · Authors · 2026-02-12
> **Response to Reviewer 3 (Part 1)**
>
> We are grateful for the reviewer’s candid and expert assessment. We recognise the reviewer’s deep familiarity with the foundational issues in TSAD, and their critique has pushed us to make the manuscript significantly more rigorous. We have addressed the specific technical corrections in the order they appear in the review:
>
> **1. "Using a range of 0 to 1... is not standard."**
>
> We have added a clarification that while not a universal standard in anomaly scoring, 0-1 normalisation of scores is a common convention.
>
> **2. "Limited applicability... You can trivially extend this scoring function... overcounting success."**
>
> We fully agree with the reviewer’s fundamental premise: success must be binary at the event level, and "overcounting" (rewarding multiple hits on the same anomaly) is a critical methodological flaw in current literature. We respectfully push back, however, on the notion that extending the UCR score to the multiple-anomaly case is "trivial" in an operational context.
>
> The UCR score constitutes a Precision@1 metric, and extending this to multiple anomalies implies a P@k metric. This requires a priori knowledge of k (the number of anomalies). In many real-world settings (particularly in the context of streaming data), k is unknown. An operator relies on thresholds, not rankings, to generate alerts. Consider if the second X in the provided example was a Y instead. Both X and Y are anomalies, but they are distinct, so the overcounting argument does not apply here. Moreover, the practical challenge of setting an appropriate threshold that balances false positives and false negatives is often underestimated.
>
> Therefore, the field requires a metric that evaluates threshold-specific outputs (where k is variable) while rigorously enforcing the reviewer’s "no overcounting of a single anomalous event" constraint. This is precisely the gap that the realistic F-score fills. It treats distinct anomalies as independent binary targets, but crucially allows for the discovery of an unknown number of events without inflating the score via redundant detections. We have revised the text to clarify that we agree with the UCR score's philosophy, but we still contend that its applicability to threshold-based evaluations is limited.
>
> **3. "Experts investigate... subsequent alerts are redundant."**
>
> We mostly addressed this in our response above, and have modified the submitted paper to caution against “overcounting success” in the case of repeated anomalies, as suggested.
>
> **4. "I think 10% is two or three orders of magnitude too high."**
>
> We agree that true anomalies are rare, and this needs to be reflected in benchmark datasets. We have revised the text to emphasise that 10% is an *absolute maximum* upper bound for a dataset to even be considered (filtering out the most egregious offenders), but that ideal benchmarks should have significantly lower density. We opted to not proffer overly stringent guidelines here, as even the best dataset in our eyes (the UCR archive) only appears to have an anomaly density of around 2.4% once the anomaly labels were padded with the suggested 100 timesteps (and around 0.8% before).
>
> **5. "The 'Magic Numbers' criticism is unfair... difference between domain heuristics and hardcoded cheating."**
>
> We appreciate the distinction the reviewer draws between generalisable heuristics (e.g. diff > 500) and specific "cheating" (e.g., diff == 8.69513`). However, we respectfully maintain our critique that the definition, as currently formulated, fails to mathematically distinguish between the two. Both examples satisfy the definition of a "one-liner using primitive operations." We argue this is a significant limitation: the definition relies on subjective human judgment to police the boundary between a "robust baseline" and "overfitting." We have rewritten **Appendix B** to acknowledge the value of using simple heuristics to intuitively demonstrate triviality, but we persist in arguing that the definition’s susceptibility to post-hoc parameter tuning limits its utility as an objective, standalone measure of difficulty.
>
> **6. "The TSB-AD archive... 'stellar work' reads as hopelessly naïve."**
>
> We appreciate the candid perspective. We have reframed the characterisation of TSB-AD as a "large-scale curation effort" while explicitly adding a cautionary note regarding concerns about label provenance.
>
> **7. "Dodgers loop... A better bit of text here would be..."**
>
> We thank the reviewer for this specific domain insight. We have updated the critique of the Dodgers Loop dataset in **Appendix C** to reflect this specific context regarding non-baseball events and inconsistent labeling.

---

> > ### Author Response · Authors · 2026-02-12
> > **Response to Reviewer 3 (Part 2)**
> >
> > **8. "I think the community should set its sights higher than the synthetic GutenTAG dataset."**
> >
> > We 100% agree that synthetic data has limitations regarding realism. We also agree that it is a relatively easy dataset, although we believe that it remains one of the better options for the time being considering the scarcity of good alternatives.
> >
> > **9. "MBA ECG dataset... is a VERY bad dataset."**
> >
> > We have accepted this correction. We have moved both datasets from "Partially recommended" to "Not recommended" and added a note that the anomalies are repetitive, meaning success on one implies success on all (overcounting).
> >
> > **10. "Fig 19 is just copied from Wu and Keogh."**
> >
> > We have modified the caption to ensure that Wu and Keogh are cited, but we should note that the figures are distinct in the sense that ours plots the distribution of all anomaly locations whereas the Wu and Keogh figure plots the rightmost anomalies (which accentuates the severity of the run-to-failure bias).
> >
> > **11. "It is not clear what this paper adds... there are no new experiments."**
> >
> > We have addressed this fundamental critique by adding an empirical study (Section 4 & Appendix A). We benchmarked 6 algorithms on UCR, Mackey-Glass, and GutenTAG using the realistic F-score. This provides quantitative evidence (rank reversals) backing the theoretical arguments.
> >
> > We thank the reviewer once again for their deep insight and feedback, and believe that our response addressed the concerns raised. Thank you for your time.

---

### Review · Reviewer_stxz · 2025-11-12

**Summary Of Contributions:**

This paper provides a critical and comprehensive examination of evaluation methodology in time-series anomaly detection. The authors argue that much of the field’s apparent progress is built on weak methodological foundations—specifically, flawed metrics, dubious datasets, and poor benchmarking practices.
The paper contributes three main elements:
- A principled critique of commonly used metrics and a proposal of two alternatives — the Realistic F1 and Realistic AUCPR — designed to better align with real-world operational use.
- A systematic review and classification of public benchmark datasets, identifying mislabeling and unrealistic anomaly densities; accompanied by visual illustrations and concrete recommendations for dataset development.
- A broader position statement advocating for renewed focus on benchmark reliability, interpretability, and efficiency, rather than continual model complexity escalation.

**Audience:**

Yes

**Audience Explanation:**

The paper addresses a central methodological issue in the machine-learning community — the misalignment between theoretical research and practical applications in time-series anomaly detection. TMLR’s audience, which values rigorous evaluation, benchmark reliability, and reproducibility, would find this work relevant.

**Broader Impact Concerns:**

No ethical or societal risks are apparent.

**Claims And Evidence:**

Yes

**Claims Explanation:**

Strengths:
1. This work provides a comprehensive literature synthesis that integrates findings from multiple critical works into a coherent narrative.
2. This paper is clear, structured, and well-written. The exposition is logically organized and figures in the appendices strongly reinforce the arguments.
3. The authors introduce concrete evaluation metrics and dataset curation principles that can directly influence community practice.

Weaknesses:
1. The new metrics (Realistic F1 / AUCPR) are conceptually well-motivated but not empirically demonstrated. Even a small-scale comparative experiment would substantiate their practical benefit.
2. The work is mainly analytical and critical, offering limited theoretical novelty.
3. The dataset quality ratings rely primarily on visual inspection and anecdotal reasoning. A quantitative validation (e.g., inter-annotator agreement, anomaly-evidence scoring) would make the critique more rigorous.

**Requested Changes:**

- Provide small-scale empirical comparisons for the proposed metrics showing how the proposed Realistic F1 and Realistic AUCPR differ from existing metrics in practice.
- Add quantitative validation of dataset assessments (e.g., inter-annotator consistency, anomaly-evidence metrics, or density statistics).

---

> ### Author Response · Authors · 2026-02-12
> **Response to Reviewer 2**
>
> We thank the reviewer for their constructive feedback, particularly regarding the need for empirical validation and quantitative profiling. We have revised the manuscript to address these points directly:
>
> **1. Empirical Validation of Metrics**
>
> We added a systematic empirical study (**Appendix A**, referenced in **Section 4**) benchmarking 6 algorithms (from baselines to deep learning models) on the UCR, Mackey-Glass, and GutenTAG datasets. We empirically demonstrate that model rankings shift significantly. For example, on the UCR dataset, the autoencoder collapses from a competitive score (0.84) under standard metrics to a poor score (0.30) under realistic metrics, while shape-based methods like the matrix profile rise to the top. We also prove that widespread metrics such as the point-adjusted F-score and event-wise F-score can be gamed using simple baseline scoring approaches.
>
> **2. Structured Artifacts & Dataset Profiling**
>
> We agree that visual inspection alone is insufficient. We have added **Table 1 (Metadata for Recommended Datasets)** in Section 5, which explicitly lists the **anomaly density (contamination)** and **event counts** for the recommended benchmarks. For the flawed datasets, we have supplemented our existing “gallery of examples” in **Appendix C** with references to the quantitative profiling in the TSB-AD archive, which statistically confirms the high anomaly densities (often >10%) criticising in our review. This combination of visual evidence and quantitative metadata ensures the critique is rigorous and verifiable.
>
> We believe these revisions robustly address the request for empirical substantiation and address the reviewer’s suggestions. Thank you once again for your time and insight.

---

### Review · Reviewer_ugaD · 2026-01-31

**Summary Of Contributions:**

**Summary:**

This paper argues that the main bottleneck in time series anomaly detection is the mismatch between common academic evaluation practices and real deployment needs. It reviews widely used metrics and explains how they can reward undesirable behavior, then proposes realistic variants that aim to count each anomalous event as detected once, while ignoring redundant true positives within the same event. The authors also discuss what makes a benchmark dataset suitable for practice, critiques several popular public datasets using these criteria, and recommends shifting more attention toward reliable benchmarking and interpretability rather than only proposing new model variants.

**Strengths:**
- I think this paper is strongly motivated by practical assumptions about how anomaly detection is used, especially human-in-the-loop review and the limited value of repeated alerts within the same anomalous segment.
- It provides a clear readable discussion of failure modes of common metrics, which can help researchers avoid misleading conclusions from standard reporting habits.
- The authors offer concrete alternatives to existing metrics rather than only criticizing prior work, which makes the paper constructive and easier to act on.
- This paper highlights dataset quality issues and provides a useful checklist for what benchmark datasets should capture, which can help guide better experimental design and dataset selection.

**Weaknesses:**
- The paper’s presentation resembles a survey or position-style review, but it lacks standard structuring elements such as taxonomy diagrams, summary tables, and structured comparisons that would make the recommendations easier to verify and reuse.
- We can see that many claims are supported mainly by qualitative reasoning and examples, without a systematic empirical study showing how model rankings change under the proposed metrics across a broad suite of algorithms and datasets.
- Importantly, the dataset critique does not include comprehensive quantitative profiling, such as distributions of anomaly lengths, anomaly densities or label consistency statistics, which limits the strength of the dataset conclusions.
- Same concern that the proposed metrics are not validated thoroughly, and the paper does not analyze sensitivity to event definitions, anomaly granularity or cost tradeoffs that commonly matter in practice.
- The scope leans heavily toward univariate benchmarks, while many real deployments are multivariate and involve correlated channels and structured alerting constraints that may change what “aligned” evaluation looks like.
- The paper calls for better benchmarking practice but does not specify an executable protocol.

**Audience:**

Yes

**Audience Explanation:**

Yes. A lot of TMLR readers work on time series or real-world ML deployment, and this paper speaks directly to a pain point that keeps coming up, we often optimize for benchmarks and metrics that do not reflect how anomaly detection is actually used. The discussion is a useful reminder of why metric choice and dataset choice matter, and the proposed realistic metrics give people a concrete starting point for rethinking how they report results.

**Broader Impact Concerns:**

1) The paper encourages more realistic benchmarking and dataset usage, which is generally positive, but please consider briefly discussing how these evaluation choices might affect safety-critical deployments.

2) If the paper recommends or curates datasets, it would be helpful to mention privacy and governance considerations.

**Claims And Evidence:**

Yes

**Claims Explanation:**

I think mostly yes, but with important caveats. The paper’s central claim that common TS anomaly detection benchmarks and metrics can be misaligned with real operational needs is well motivated, and supported by clear reasoning and concrete examples. The discussion of metric failure modes is convincing at a high level, and the dataset critiques point to real issues that the community should take seriously. That said, several of the stronger takeaways would be more compelling with a more systematic empirical study. In particular, I would like to see broader quantitative evidence showing how model rankings change under the proposed metrics, and clearer validation of when the paper’s assumptions match typical deployment settings. So the evidence is clear for the main message, but some conclusions still feel more suggestive than fully demonstrated.

**Requested Changes:**

1) Could you please add a more systematic empirical study around the proposed metrics? In particular, it would really help to evaluate a reasonably broad set of representative TSAD methods on multiple widely used benchmarks, and then show how the rankings shift under standard metrics versus realistic F1 and realistic AUCPR. Right now the motivation is clear, but it is hard to judge impact without seeing how conclusions change in practice.

2) Please consider adding a couple of structured artifacts that make the paper easier to audit and reuse. This is also critical for me. A dataset table in the same spirit would also be valuable.

3) It would strengthen the dataset discussion if you could include basic quantitative profiling for the main benchmarks you comment on. This is important but not strictly required for acceptance.

4) Please also clarify the scope of the realistic setting and discuss edge cases a bit more. It would be helpful to explain when the one reward per event assumption matches typical operations and when it might not.

---

> ### Author Response · Authors · 2026-02-12
> **Response to Reviewer 1**
>
> We thank the reviewer for their thoughtful and constructive feedback. We appreciate the recognition of the paper’s practical motivation and the value of our critique regarding metric and dataset failure modes.
>
> We have revised the manuscript to address the reviewer's requests, specifically adding a new systematic empirical study and quantitative profiling of datasets. Below is a detailed account of the changes:
>
> **1. Empirical validation of metrics ("Systematic empirical study... showing how model rankings change")**
>
> We agree that qualitative reasoning alone was insufficient. We have added a comprehensive empirical study (now **Appendix A** and referenced in **Section 4**) benchmarking 6 distinct anomaly scoring algorithms (ranging from simple baselines to deep learning models) on three major datasets (UCR, Mackey-Glass, GutenTAG).
>
> We now explicitly demonstrate how model rankings shift. For instance, on the UCR dataset, the autoencoder appears competitive under the standard point-adjusted F1 with a score of 0.84, but collapses to 0.30 under the realistic F1.
>
> We also included “Random” and “Flag All” baselines to empirically prove that other metrics award "state-of-the-art" scores to these trivial baselines, whereas the recommended realistic F-score correctly scores them near zero.
>
> **2. Structured Artifacts & Dataset Profiling ("Quantitative profiling... dataset table")**
>
> To make the paper more audit-friendly, we have added **Table 1 (Metadata for Recommended Datasets)** in Section 5. This table quantitatively profiles the anomaly counts, contamination levels, and lengths for the recommended benchmarks. For the remaining datasets, we have added references to the comprehensive profiling available in the TSB-AD archive to avoid redundancy, while providing a visual "gallery of failure modes" in **Appendix C** that profiles the specific flaws (dubious labels, high density) of the datasets criticised in the text.
>
> **3. Clarification of "Realistic" Scope & Edge Cases**
>
> We have refined **Section 3.2** to clarify that the "one reward per event" assumption is primarily targeted at human-in-the-loop monitoring contexts where alarm fatigue is a critical bottleneck. We now explicitly acknowledge edge cases such as processes where duration-aware metrics might still be necessary.
>
> **4. Broader impact**
>
> We have added a brief discussion in the conclusion on how these evaluation choices might affect safety-critical deployments. We highlight the safety risks of optimising for inflated metrics in domains such as healthcare. It is argued that "optimistic" reporting can lead to the deployment of models that appear reliable on paper but fail to provide timely, actionable alerts in practice, potentially endangering patients or systems.
>
> We believe these changes significantly strengthen the evidence base of the paper and make the recommendations more actionable for the TMLR community. Thank you once again for your detailed feedback.

---

### Decision · Action_Editor_cjUB · 2026-04-03

**Recommendation:** Accept as is

**Audience:**

Yes

**Audience Explanation:**

time series anomaly detection is certainly an important and popular research topic in machine learning which will attract TMLR audience.

**Claims And Evidence:**

Yes

**Claims Explanation:**

This paper pointed out that the main bottleneck in time series anomaly detection is the mismatch between common academic evaluation practices and real deployment needs. It reviews widely used metrics and explains how they can reward undesirable behaviour, then proposes realistic variants that aim to count each anomalous event. The authors also discuss what makes a benchmark dataset suitable for practice, critiques several popular public datasets using these criteria, and recommends shifting more attention toward reliable benchmarking and interpretability rather than only proposing new model variants. The writing is clear and the critique is constructive, and the proposed event-aware metric variants are a reasonable direction.

Therefor, the paper addressed an important limitation of the existing methods and also proposed  event-aware metric variants to address this limitation.